# Pharmacological Ascorbate Elicits Anti-Cancer Activities against Non-Small Cell Lung Cancer through Hydrogen-Peroxide-Induced-DNA-Damage

**DOI:** 10.3390/antiox12091775

**Published:** 2023-09-18

**Authors:** Kittipong Sanookpan, Naphat Chantaravisoot, Nuttiya Kalpongnukul, Chatchapon Chuenjit, Onsurang Wattanathamsan, Sara Shoaib, Pithi Chanvorachote, Visarut Buranasudja

**Affiliations:** 1Department of Pharmacology and Physiology, Faculty of Pharmaceutical Sciences, Chulalongkorn University, Bangkok 10330, Thailand; kittipong.s@nabsolute.co.th (K.S.); onsurang.w@gmail.com (O.W.); sarashoaibahmed@gmail.com (S.S.); pithi.c@chula.ac.th (P.C.); 2Nabsolute Co., Ltd., Bangkok 10330, Thailand; 3Department of Biochemistry, Faculty of Medicine, Chulalongkorn University, Bangkok 10330, Thailand; naphat.c@chula.ac.th (N.C.); chatchapon.cj@gmail.com (C.C.); 4Center of Excellence in Systems Microbiology, Faculty of Medicine, Chulalongkorn University, Bangkok 10330, Thailand; 5Center of Excellence in Systems Biology, Faculty of Medicine, Chulalongkorn University, Bangkok 10330, Thailand; nuttiya.nkal@gmail.com; 6Research Affairs, Faculty of Medicine, Chulalongkorn University, Bangkok 10330, Thailand; 7Center of Excellence in Cancer Cell and Molecular Biology, Faculty of Pharmaceutical Sciences, Chulalongkorn University, Bangkok 10330, Thailand; 8Center of Excellence in Natural Products for Ageing and Chronic Diseases, Faculty of Pharmaceutical Sciences, Chulalongkorn University, Bangkok 10330, Thailand

**Keywords:** pharmacological ascorbate, anti-cancer, non-small cell lung cancer, DNA damage, pro-oxidant, adjuvant, vitamin C, oxidative distress, ascorbic acid

## Abstract

Non-small cell lung cancer (NSCLC) poses a significant global health burden with unsatisfactory survival rates, despite advancements in diagnostic and therapeutic modalities. Novel therapeutic approaches are urgently required to improve patient outcomes. Pharmacological ascorbate (P-AscH^−^; ascorbate at millimolar concentration in plasma) emerged as a potential candidate for cancer therapy for recent decades. In this present study, we explore the anti-cancer effects of P-AscH^−^ on NSCLC and elucidate its underlying mechanisms. P-AscH^−^ treatment induces formation of cellular oxidative distress; disrupts cellular bioenergetics; and leads to induction of apoptotic cell death and ultimately reduction in clonogenic survival. Remarkably, DNA and DNA damage response machineries are identified as vulnerable targets for P-AscH^−^ in NSCLC therapy. Treatments with P-AscH^−^ increase the formation of DNA damage and replication stress markers while inducing mislocalization of DNA repair machineries. The cytotoxic and genotoxic effects of P-AscH^−^ on NSCLC were reversed by co-treatment with catalase, highlighting the roles of extracellular hydrogen peroxide in anti-cancer activities of P-AscH^−^. The data from this current research advance our understanding of P-AscH^−^ in cancer treatment and support its potential clinical use as a therapeutic option for NSCLC therapy.

## 1. Introductions

Lung cancer stands as the top rank in cancer-related mortality globally, causing approximately 1.8 million fatalities, and ranks as the second most commonly diagnosed cancer with around 2.2 million cases reported [1]. Non-small cell lung cancer (NSCLC) represents the predominant subtype, accounting for approximately 85% of all of lung cancer cases [2]. Despite advancements in diagnostic and therapeutic technologies, the majority of patients are still diagnosed at advanced stages, resulting in unsatisfactory treatment outcomes with 5-year survival rates below 10% [2,3]. Consequently, there is an urgent demand for a novel therapeutic approach that can enhance effectiveness of treatment and prolong survival of patients affected by this malignancy.

Ascorbate, commonly known as vitamin C, is an essential micronutrient that plays crucial roles in diverse biochemical processes. Depending on its concentration, ascorbate can exhibit either antioxidant or pro-oxidant characteristics. At physiological levels (around 40–80 μM in plasma), ascorbate primarily serves as a donor antioxidant. This property is essential as it enables ascorbate to effectively scavenge reactive oxygen species and functions as co-factors for various enzymes [4]. When presented at pharmacological concentration (plasma ascorbate levels of 10–20 mM), known as pharmacological ascorbate (P-AscH^−^), ascorbate can manifest its pro-oxidant activities through generation of a high flux of H_2_O_2_. This pro-oxidant feature of P-AscH^−^ is believed to be the key contributor of anti-tumor activities of P-AscH^−^ [5,6,7,8,9,10,11,12,13].

The clinical potential of high-dose ascorbate in cancer treatment was first reported by Ewan Cameron and Linus Pauling in the 1970s [14]. However, two randomized controlled trials in the 1980s failed to confirm the initial observations on patients with advanced malignant diseases, despite using similar doses of ascorbate [15,16]. Consequently, the medical and scientific communities abandoned the clinical utilities of high-dose ascorbate in cancer therapy. One key distinction between the initial and subsequent trials lies in the route of ascorbate administration. The study from the 1970s employed a combination of oral and intravenous ascorbate, whereas the later studies administered ascorbate solely orally. Subsequent pharmacokinetic studies revealed that pharmacological concentration of ascorbate (millimolar concentration) can only be achieved through an intravenous infusion, while oral administration results in significantly lower bioavailability of ascorbate (micromolar concentration) due to limitations in gastrointestinal absorption, tissue transport and renal excretion [17,18,19,20,21]. This improved understanding of ascorbate’s pharmacokinetic profiles reignited interest in its clinical benefits.

Over the past few decades, a growing number of both preclinical and clinical investigations highlighted the therapeutic potential of P-AscH^−^ in cancer therapies. Results from cell culture and animal studies show that P-AscH^−^ exhibits selectivity toward various cancer cells without harming non-malignant cells [5,7,22,23,24]. These selective actions might be attributed to the differential capacity of cells to eliminate H_2_O_2_. Cancer cells generally exhibit a significantly lower capacity to remove H_2_O_2_ compared to normal cells [5,23]. Consequently, P-AscH^−^ exhibits preferential cytotoxic effects on cancer cells. Furthermore, recent findings from both preclinical and clinical studies provided an insight into the potential of P-AscH^−^ to enhance the efficacy of standard-of-care treatments (e.g., chemotherapy [25,26,27,28], radiotherapy [7,29,30], and chemoradiotherapy [31,32,33]), while concurrently mitigating their associated adverse effects. Taken together, these distinctive properties of P-AscH^−^ position it as a promising candidate for management of cancer.

Although researchers evaluated potential utilities of P-AscH^−^ in cancer treatments for over five decades, the specific mechanism by which its anti-cancer effects work against NSCLC remains unclear. In our current investigation, NCI-H23 (H23; RRID: CVCL_1547), NCI-H292 (H292; RRID: CVCL_0455), and NCI-H460 (H460; RRID: CVCL_0459) cell lines were used as a representative cell culture model for NSCLC. Our results highlight the crucial role of extracellular H_2_O_2_ formation in the cytotoxicity of P-AscH^−^ against NSCLC. We demonstrated that the high-flux of H_2_O_2_ generated by P-AscH^−^ leads to the induction of intracellular oxidative distress; oxidative DNA damage; disruption of DNA repair processes; perturbation of cellular bioenergetics; and ultimately stimulation of apoptotic cell death across all NSCLC cell lines tested. Additionally, our findings shed light on the significance of DNA damage as a potential key contributor to the therapeutic effects of P-AscH^−^ against NSCLC.

## 2. Materials and Methods

### 2.1. Materials

L-ascorbic acid (cat. No. A92902), adenosine 5′-triphosphate disodium salt hydrate (ATP; cat. No. A26209), catalase from bovine liver (cat. No. C9322), crystal violet, dimethyl sulfoxide (DMSO), 2′,7′-dichlorofluorescin diacetate (DCFH-DA; cat. No. D6883), L-glutathione reduced (GSH; cat. No. G4251), Immobilon Western chemiluminescent HRP substrate (cat. No. P90719), methylthiazolyldiphenyl-tetrazolium bromide (MTT; cat. No. M5655), and paraformaldehyde were purchased from MilliporeSigma (Burlington, MA, USA). Annexin V/propidium iodide staining kit was purchased from ImmunoTools (Friesoythe, Germany). GSH-Glo Glutathione assay (cat. No. V6911), CellTiter Glo luminescent cell viability assay (cat. No. G7571), and NAD/NADH Glo assay (cat. No. G9071) were purchased from Promega (Madison, WI, USA). Pierce BCA protein assay kit (cat. No. 23225), cell culture media, and supplements were obtained from Thermo Fisher Scientific (Waltham, MA, USA). Radioimmunoprecipitation assay (RIPA) buffer (cat. No. 9806), protease/phosphatase inhibitor cocktail (100×; cat. No. 5872), primary antibodies, and secondary antibodies were bought from Cell Signaling Technology (Danver, MA, USA). Gemcitabine hydrochloride (cat. No. G0367) and docetaxel (cat. No. D4102) were purchased from Tokyo Chemical Industry (Tokyo, Japan).

### 2.2. Cell Culture

Human NSCLC cell lines, including H23, H292, H460, and A549 (RRID: CVCL_0023) cells were obtained from the American Type Culture Collection (ATCC; Manassas, VA, USA). The H23, H292, and H460 cells were cultured in RPMI-1640, supplemented with penicillin (100 unit/mL/streptomycin 100 μg/mL) and 10% fetal bovine serum (FBS), while A549 cells were cultured in DMEM, complemented with penicillin (100 unit/mL)/streptomycin 100 μg/mL), along with 10% FBS. All these cell lines were maintained in a controlled environment at 37 °C with a humidified atmosphere containing 95%air/ 5% CO_2_.

### 2.3. Treatments with P-AscH^−^

Stock solutions of ascorbate (1.00 M, pH 7.00) were made under nitrogen gas. The concentration of prepared solutions were determined with a Cary 60 UV/vis spectrophotometer (Agilent Technologies, Santa Clara, CA, USA) at 265 nm with an absorption coefficient (ε) of 14,500 M^−1^ cm^−1^ [34]. The stock solutions were stored in a borosilicate glass tube with minimum headspace.

The generation of H_2_O_2_ through oxidation of P-AscH^−^ is influenced by the pH of culture media. Hence, to minimize variations in H_2_O_2_ flux among different studies, it is necessary to exchange the medium before P-AscH^−^ treatments [34]. Following the replacement with fresh serum-free medium, NSCLC cells were treated with ascorbate at 37 °C for 1 h and the subsequent biological consequences were observed as indicated.

### 2.4. Treatments with Extracellular Catalase

Extracellular catalase was used to observe the impacts of extracellular H_2_O_2_ on the anti-cancer effects of P-AscH^−^. The stocks of catalase were made in water, sterilized with 0.22 μm filter, and kept at 4 °C. Cells were incubated with catalase (200 U/mL) immediately before exposure to P-AscH^−^.

### 2.5. Treatments with Gemcitabine or Docetaxel

The stock solutions of gemcitabine and docetaxel were prepared in DMSO and the aliquots were kept at −20 °C. Cells were subjected to treatment with gemcitabine or docetaxel under indicated conditions. The control groups were exposed to an equivalent volume of DMSO (serving as the vehicle control).

### 2.6. MTT Viability Assay

Cells were seeded into a 96-well culture plate (10,000 cells/well) and cultured for 24 h. After the designated treatments, cells were rinsed twice with phosphate buffer solution (PBS; pH 7.0) and allowed to recover in complete medium for an additional 24 h. Subsequently, the viability of cells was assessed with MTT assay. Briefly, the cells were washed twice with PBS and incubated with MTT solution (1 mg/mL in serum-free medium; 200 μL) in a dark environment for 3 h. Following this incubation period, the MTT solution was carefully discarded, and the resulting insoluble formazan crystals were solubilized with DMSO (200 μL). Absorbance readings of formazan solution were taken at wavelength of 570 nm with CLARIOStar microplate reader (BMG Labtech, Ortenberg, Germany). The effective dose of P-AscH^−^ that resulted in 50% reduction in viability (ED_50_) for each cell line was further estimated by analyzing the dose–response relationship with GraphPad Prism 10.0.0 (GraphPad Software, San Diego, CA, USA).

### 2.7. Clonogenic Survival Assay

Cells were initially seeded into a 24-well culture plate (20,000 cells/well) and cultured for a period of 24 h. After treatments, the cells were washed twice with PBS and gently harvested through trypsinization. Cell counting was conducted with a hemocytometer, and cells were reseeded in triplicate into a 6-well culture plate at 500 cells in 2.0 mL of complete medium in each well. Subsequently, the cells were cultured for approximately 7–14 days. Surviving colonies were fixed with 4% paraformaldehyde for 20 min and stained with 2% crystal violet for 10 min. A colony was characterized as a group consisting of a minimum of 50 cells.

### 2.8. Determination of Cellular Redox Status with DCFH-DA

Cells were seeded into a 24-well culture plate (20,000 cells/well) and cultured for 24 h. Prior to the treatments, cells were stained with DCFH-DA (5 μM in serum free RPMI-1640 for 30 min). Following specified treatments, cells were washed twice with ice-cold PBS, gently collected with trypsinization, and resuspended in ice-cold PBS. The oxidation of DCFH-DA was subsequently quantified using a Guava easyCyte HT flow cytometer (MilliporeSigma, Burlington, MA, USA) at an excitation wavelength of 485 nm and an emission wavelength of 530 nm. The mean fluorescence intensity was calculated using Guava Incyte software version 2.7 (MilliporeSigma, Burlington, MA, USA).

### 2.9. Measurement of Intracellular GSH

DCFH-DA-based assay has several artifacts in studying the redox biology of cells, such as interferences from redox active metals, interaction with heme protein, and artifactual amplification of the fluorescent signal from the futile redox cycle [35]. To overcome these limitations, we further validate the findings obtained from DCFH-DA studies by determination of the intracellular GSH levels.

Cells were seeded into 24-well culture plate (20,000 cells/well) and cultured for 24 h prior to exposure to experimental conditions. Following treatments, cells were rinsed twice with PBS, harvested with trypsin, and resuspended in ice-cold PBS. The intracellular levels of GSH were evaluated with GSH-Glo glutathione assay following the manufacturer’s protocol. This assay utilizes luminescence to quantify amounts of intracellular GSH. It is based on the principle that glutathione S-transferase (GST) converts luminogenic substrate into luciferin in the presence of GSH. Subsequently, the resulting luciferin couples with luciferase, leading to the generation of a luminescent signal. Hence, the intensity of a luminescent signal is directly proportional to the amount of intracellular GSH in the tested samples.

Briefly, 10,000 cells in 50 μL of PBS were introduced into experimental wells of a 96-well white microplate with opaque-bottom. The suspended cells were then incubated with an equal volume of GSH-Glo reagent for 30 min. Afterward, 100 μL of luciferin detection reagent was added and further incubated for 15 min. Once the incubation was completed, the luminescent signal was recorded with a CLARIOStar microplate reader. The concentrations of GSH were estimated by the standard curve and subsequently converted to intracellular concentration per cells based on the cell number. Notably, the GSH-Glo reagent mainly consists of a luminogenic substrate and glutathione S-transferase (GST), while the luciferin detection reagent contains firefly luciferase.

### 2.10. Measurement of Intracellular ATP

The conditions for cell culture and treatment protocol were conducted in a similar manner as described in GSH measurement. Following indicated treatments, the intracellular levels of ATP were assessed with CellTiter-Glo luminescent cell viability assay as previously described [36]. This assay relies on the activity of luciferase, which produces the luminescent signal in the presence of ATP. The intensity of a luminescent signal directly correlates with the levels of ATP in the tested samples.

Briefly, 10,000 cells suspended in 100 μL of ice-cold PBS were added into wells of a 96-well white microplate with an opaque bottom. An equal volume of CellTiter-Glo solution was then mixed with cell suspension to induce cell lysis and initiate the luminescence reaction. Following a 10-min incubation period, the luminescent signals were recorded with a CLARIOStar microplate reader. The amounts of ATP were determined using a standard curve and then further converted to intracellular ATP per cell with the cell number.

### 2.11. Measurement of NAD^+^ Pools

The conditions for cell culture and treatment protocol were similar to those conducted in quantification of intracellular GSH and ATP. Following the treatments, the intracellular pools of NAD^+^ were quantified using NAD/NADH-Glo assay following the recommended guideline from the manufacturer. This luminescent-base assay enables the measurement of total oxidized and reduced nicotinamide adenine dinucleotides (NAD^+^ and NADH, respectively). The underlying principle of this approach relies on the function of the NAD cycling enzyme, which transforms NAD^+^ to NADH. In the presence of NADH, reductase enzyme converts pro-luciferin reductase substate into luciferin. Subsequently, recombinant luciferase catalyzes the conversion of luciferin, resulting in the formation of a luminescent signal. Hence, the intensity of the luminescent signal is directly proportional to levels of NAD^+^ pool (NAD^+^ and NADH) in the tested sample.

Briefly, cells suspensions (10,000 cells in 50 μL ice-cold PBS) were added into wells of a 96-well white microplate with an opaque bottom. An equal volume of NAD/NADH-Glo detection reagent was then mixed with cell suspension. Following a 30-min incubation period, the luminescent signals were quantified with a CLARIOStar microplate reader. The intracellular level of the NAD^+^ pool in the sample was reported relatively to untreated control.

### 2.12. Characterization of Cell Death

Annexin V and propidium iodide (PI) co-staining is a commonly method for distinguishing between apoptotic and necrotic cell death. Cells were plated into a 24-well culture plate (20,000 cells/well) and cultured for 24 h. Following treatments, cells were washed twice with PBS and allowed to recover for 24 h. Subsequently, cells were rinsed twice with ice-cold PBS, collected with trypsinization, and suspended in binding buffer. The cells were then incubated with annexin V/PI for 30 min. Apoptotic and necrotic cells were quantified using a Guava easyCyte HT flow cytometer. The mean fluorescence intensity was analyzed with Guava Incyte software.

### 2.13. Western Blot Analysis

Following indicated treatments, protein extraction was performed with a RIPA buffer supplemented with a cocktail of protease and a phosphatase inhibitor. The Pierce BCA protein assay kit was used to determine the contents of protein in the sample. Equal amounts of protein lysates were separated by SDS-PAGE and transferred onto a PVDF membrane through electroblotting. Subsequently, membranes were blocked with 5% non-fat milk in TBS-T for 1 h at 25 °C, followed by overnight hybridization with a specific primary antibody at 4 °C. After incubation with the primary antibody, the membrane was washed three times with TBS-T and incubated with suitable secondary antibodies conjugated with horseradish peroxidase (1:2000) for 1 h at 25 °C. The chemiluminescent signal was developed with Immobilon Western chemiluminescent HRP substrate. The densitometric analysis of the protein bands was conducted with ImageJ software version 1.53t (U.S. National Institutes of Health, Bethesda, MD, USA). To ensure equal loading, GAPDH was used as an internal loading control.

The following primary antibodies were used at 1:1000 dilution; anti-Chk1 (cat. No. 2360); anti-p-Chk1 (cat. No. 2348); anti-RPA2 (cat. No. 35869); anti-p-RPA2 (cat. No. 10148); anti-γ-H2AX (cat. No. 9718); anti-PARP1 (cat. No. 9532); anti-Bcl-2 (cat. No. 4223); anti-Mcl-1 (cat. No. 94296); and anti-GAPDH (cat. No. 5174). The anti-rabbit IgG, HRP-conjugated antibody (cat. No. 7074) and anti-mouse IgG, and HRP-conjugated antibody (cat. No. 7076) were used as the secondary antibody at 1:2000.

The uncropped original Western blots were illustrated in Appendix A.

### 2.14. Immunofluorescence Staining and Confocal Microscopy

H460 cells were cultured in an 8-well Lab-Tek chamber slide (Thermo Fisher Scientific). Cells were treated with P-AscH^−^ (4 mM), bovine catalase (200 U/mL), or co-treated with P-AscH^−^ and catalase for 1 h. At 6 h post-treatment, cells were fixed for 10 min with 4% paraformaldehyde in PBS. After that, cells were permeabilized by 0.2% Triton X-100 and blocked with PBS buffer containing 5% bovine serum albumin. Primary antibodies used for immunofluorescence staining included anti-γ-H2AX (cat. No. 9718), anti-RAP80 (cat. No. 14466), and anti-BRCA1 (cat. No. 14823) from cell signaling. The Zenon Rabbit IgG labeling kit (Thermo Fisher Scientific) was used to label primary antibodies. Cells were incubated with fluorescently labeled antibodies for 1 h. Nuclei were stained with 4′,6′-diamidino-2-phenylindole (DAPI). The slide was mounted using ProLong Antifade Mountant (Thermo Fisher Scientific). Confocal microscopy was performed using an LSM800 with an Airyscan confocal microscope (Zeiss, Jena, Germany). Details regarding the parameters for confocal microscopy are listed in Appendix A.

To quantify the levels of cytoplasmic γ-H2AX per cell, colocalization analysis between γ-H2AX and DAPI was performed using ZEISS ZEN 3.3 (blue edition; Zeiss) software. In each image, the pixel intensities of the γ-H2AX channel were plotted against those from the DAPI channel. Non-colocalizing pixels of γ-H2AX were specifically chosen to represent the cytoplasmic γ-H2AX within the cells. These values were subsequently normalized by total cell count per image. The resulting fold changes, relative to the untreated control, were then calculated. Importantly, the quantitation of cytoplasmic γ-H2AX was computed from a minimum of 30 cells for each treatment condition.

### 2.15. Calculation of Combination Index

For the assessment of potential synergistic interaction between P-AscH^−^ and chemotherapeutic drugs (gemcitabine or docetaxel), the combination index (CI) was calculated using the method of Chou and Talalay [37]. The experiments were conducted with six drug concentrations at a consistent dose ratio. The obtained data were calculated and analyzed with Compusyn software version 1.0.1 (Combosyn, Inc., Paramus, NJ, USA).

### 2.16. Statistical Analysis

All statistical analyses were performed using GraphPad Prism. One-way analysis of variance (ANOVA) with Tukey’s post hoc was used to calculate statistical differences. All means were calculated from at least three independent experiments. The data were expressed as mean ± standard error of the mean (SEM), unless stated otherwise.

## 3. Results

### 3.1. Production of H_2_O_2_ Is a Key Factor for Anti-Cancer Activities of P-AscH^−^ against NSCLC

In this current study, we utilized three NSCLC cell lines, namely H292, H460, and H23 as in vitro models of NSCLC. The concentrations of P-AscH^−^ used in our cell culture experiments (0.25–6 mM) are feasible in clinical practice, as intravenous infusion of high-dose ascorbic acid in humans can achieve plasma concentrations of ascorbate up to 32 mM [38,39,40]. Data from MTT viability assay demonstrated that P-AscH^−^ dose-dependently killed all three NSCLC lines (Figure 1A). The ED_50_ of P-AscH^−^ (dosage of P-AscH^−^ that causes 50% reduction in cell viability) was found to be relatively comparable for each NSCLC cell line (Figure 1B). Additionally, we observed inhibitory effects of P-AscH^−^ on the clonogenic survival of NSCLC cells. Consistent with MTT assay, treatments with P-AscH^−^ mitigated clonogenicity of NSCLC cells in a dose-dependent fashion across all NSCLC-tested cells (Figure 1C,D). These findings suggest that P-AscH^−^ possesses anti-cancer activities against NSCLC.

Activating mutations in *Kristen Rat Sarcoma* viral oncogene (*KRAS*) represents the most prevalent genetic aberrations in NSCLC patients, manifesting at a frequency around 25–30% of cases [41,42]. Among the *KRAS* mutation, the KRAS p.G12C stands out as the most predominant subtype in NSCLC [43]. In our current study, we evaluated the cytotoxic impact of P-AscH^−^ on NSCLC cell lines harboring distinct *KRAS* genetic background; H292 with wild-type KRAS; H23 with KRAS^G12C^; and H460 with KRAS^Q61H^ (Appendix A). Notably, all three cell lines exhibit wild-type EGFR status (Appendix A). Through both MTT assay and clonogenic survival assays, our results demonstrate that the anti-cancer efficacy of P-AscH^−^ remains consistent across cell lines, regardless of their respective KRAS profiles. Hence, the results from this current report suggest that the KRAS status of NSCLC may not significantly influence the responsiveness of P-AscH^−^.

The pro-oxidant properties of P-AscH^−^ were proposed to be the major factor responsible for its anti-cancer activities [5,22,23,24,44]. To examine the contribution of pro-oxidant activities of P-AscH^−^, NSCLC cells were exposed to P-AscH^−^ for 1 h and the levels of intracellular GSH were immediately measured after treatment. We demonstrated that the intracellular levels of GSH of NSCLC were instantly reduced in a dose-dependent manner following P-AscH^−^ treatments (Figure 2A), indicating the impact of P-AscH^−^‘s pro-oxidant actions. Treatment with P-AscH^−^ at 4 mM rapidly depleted the pool of intracellular GSH to approximately 30% of basal levels (Figure 2A). These reductions in GSH were paralleled to information obtained from DCFH-DA-based assays. The quantitative results from flow cytometric analysis demonstrated that the fluorescence intensity for oxidation of DCFH was substantially increased by 15–22 times after P-AscH^−^ treatment (Figure 2B,C). Additionally, fluorescence microscopy with a DCFH-DA probe illustrated enhanced DCFH oxidation following exposure to P-AscH^−^ (Appendix A). These results strongly support the association between P-AscH^−^’s pro-oxidant activities and its anti-cancer effects against NSCLC.

Generation of extracellular H_2_O_2_ was proposed as a potential mechanism for induction of oxidative distress by P-AscH^−^ [5,13,22,23,24,44,45]. To investigate this possibility, we used catalase as a tool to examine the contribution of extracellular H_2_O_2_ to the anti-cancer properties of P-AscH^−^ [23]. Co-treatment with catalase significantly suppressed oxidation of DCFH by P-AscH^−^ in all tested cell lines (Figure 2B,C; Appendix A). Hence, the induction of oxidative distress by P-AscH^−^ is possibly due to the production of extracellular H_2_O_2_. In addition, clonogenic properties of NSCLC cells were preserved with catalase co-treatment (Figure 2D,E). These preventive effects of extracellular catalase against P-AscH^−^ cytotoxicity were also observed in other NSCLC cell lines, A549 cells (Appendix A). These results emphasize the significance of the generation of extracellular H_2_O_2_ as a crucial factor contributing to the cytotoxicity of P-AscH^−^ against NSCLC.

### 3.2. P-AscH^−^ Causes Reduction in Intracellular ATP and NAD^+^ of NSCLC Cells through H_2_O_2_ Formation

The highly nucleophilic properties of DNA make it vulnerable to attack by reactive oxygen species, including hydrogen peroxide, during oxidative imbalance condition [46]. In this study, we aimed to investigate whether DNA is a potential target for P-AscH^−^ by examining (i) alterations in cellular bioenergetics; and (ii) changes in DNA damage response machineries.

During genotoxic stress, there is a significant increase in consumption of energy for activation of the DNA damage response system [47]. Our bioenergetic studies showed that P-AscH^−^ treatments resulted in a rapid, dose-dependent depletion of intracellular levels of ATP (Figure 3A) and NAD^+^ pools (Figure 3B) of H23, H292, and H460 cells. Exposure to P-AscH^−^ at 4 mM caused a remarkable decrease in intracellular ATP levels (approximately 6–17% of untreated NSCLC cells) and NAD^+^ pools (about 6–22% of untreated NSCLC cells) (Figure 3A,B). These severe reductions in both intracellular ATP (Figure 3C) and NAD^+^ pools (Figure 3D) were significantly attenuated in the presence of extracellular catalase. The protective roles of extracellular catalase against the disturbance in bioenergetics equilibrium following exposure to P-AscH^−^ were also evident in A549 cells (Appendix A). These results strongly indicate that P-AscH^−^ disrupts bioenergetic homeostasis of NSCLC cells via generation of extracellular H_2_O_2_.

### 3.3. DNA Damage Is a Potential Mechanism for Anti-Cancer Activities of P-AscH^−^ against NSCLC

To gain further insight into the genotoxic activities of P-AscH^−^ against NSCLC cells, we investigated several key phenomena associated with DNA damage response following P-AscH^−^ treatments. Rapid phosphorylation at serine 139 of histone variant H2AX, known as γ-H2AX, is an early cellular response to DNA damage. The γ-H2AX is a critical prerequisite for stimulation of DNA damage response cascade. Hence, formation of γ-H2AX is widely used as a biomarker for genotoxic stress [48]. A time-course study using Western blot assay revealed that formation of γ-H2AX increased immediately upon 1 h treatment with P-AscH^−^ on all NSCLC cells and persisted for up to 24 h (Figure 4A,B). These results suggest that the oxidative DNA damage could be a contributing factor for cytotoxicity of P-AscH^−^ on NSCLC.

Replication protein A (RPA) is a heterotrimeric protein complex that plays an essential role in the maintenance of genetic integrity. The RPA complex is composed of three subunits, including RPA1, RPA2, and RPA3. Within the RPA2 subunit, there are multiple residues, including serine 33, that undergo phosphorylation by upstream kinases in response to genotoxic stress. These hyperphosphorylations on RPA2 are important for recruitment and activation of downstream genome maintenance mediators [49,50]. We observed a time-dependent reduction in RPA2 expression across all NSCLC cells following treatments with P-AscH^−^. At 24 h post-treatment, the expressions of RPA2 were downregulated to less than 40% (Figure 4A,B). Moreover, phosphorylation of RPA2 at serine 33 were observed in all NSCLC cells at 6 h after P-AscH^−^ treatments. The signals of phosphorylated RPA2 were mostly disappeared within 24 h (Figure 4A,B). Altogether, this information strongly suggests that exposure to P-AscH^−^ could induce genotoxic stress and disrupt DNA damage response cascade of NSCLC cells.

Checkpoint kinase 1 (Chk1) is a crucial effector of the DNA damage response system. Phosphorylation of Chk1 by upstream transducer is an essential event for maintenance of genomic stability upon replication stress [51]. We found that exposure to P-AscH^−^ caused a substantial reduction in Chk1 levels in NSCLC cells. The levels of Chk1 were decreased to approximately 50% immediately after treatment, and continuously downregulated to about 37% and 24% at 6 h and 24 h post-treatments, respectively (Figure 4A,B). Furthermore, the formation of phosphorylation at serine 345 of Chk1 was detected in all NSCLC cells at 6 h following treatment with P-AscH^−^ (Figure 4A,B). The phosphorylation of Chk1 following treatments with P-AscH^−^ is occurred in parallel with generation of phosphorylated RPA2. These results highlight the contribution of genotoxicity in the anti-cancer activities of P-AscH^−^.

### 3.4. Induction of DNA Damage Following P-AscH^−^ Treatment Is Due to Generation of Extracellular H_2_O_2_

Results from time-course studies show that substantial alterations in DNA damage response cascade were detected at 6 h following treatments (Figure 4). Hence, this period was chosen for further observation of DNA damage induced by P-AscH^−^. At 6 h after challenge with P-AscH^−^, the formation of DNA damage and replication stress markers (γ-H2AX, p-Chk1, and p-RPA2) in NSCLC cells were markedly suppressed with extracellular catalase co-treatment. Moreover, catalase completely abrogated the downregulation of key regulators of DNA damage response (Chk1 and RPA2) following P-AscH^−^ treatments across all NSCLC tested cells (Figure 5). These results indicate that the genotoxicity of P-AscH^−^ against NSCLC could be dependent on the generation of extracellular H_2_O_2_.

### 3.5. Pharmacological Ascorbate Disrupts Accumulation of DNA Repair Machineries at Damage Sites

In the early phase of DNA damage, the presence of γ-H2AX plays a pivotal role in attracting downstream DNA repair machineries, such as breast cancer 1 (BRCA1) and receptor-associated protein 80 (RAP80) to the specific sites where DNA breaks occurred. The accumulation of these DNA repair factors at the damaged sites is vital for the amplification and subsequent activation of DNA damage responses [52,53]. Confocal microscopic studies on H460 cells demonstrated that P-AscH^−^ disrupted the accumulation of DNA repair factors at damaged sites. We found that a formation of γ-H2AX and localization of DNA repair proteins (BRCA1 and RAP80) were not predominantly confined to the nucleus, but also more dispersed in the cytoplasm after P-AscH^−^ treatments (Figure 6A). To quantitatively assess these miscolocalization phenomena, we chose to measure cytoplasmic γ-H2AX levels, given its pivotal roles in recruiting and accumulating downstream repair factors at damage sites [48]. Our quantitative analysis demonstrated that treatments with P-AscH^−^ led to an approximately 3.3-fold increase in cytoplasmic γ-H2AX levels, as compared to untreated control (Figure 6B and Appendix A). These findings suggest that P-AscH^−^ may potentially attack the nuclear membrane of cancer cells, leading to a leakage of DNA repair components and subsequent impairment of DNA repair processes. Remarkably, these cytoplasmic accumulations of DNA repair factors caused by P-AscH^−^ were mitigated by co-treatment with extracellular catalase (Figure 6A,B). These results strongly highlight the importance of pro-oxidant properties of P-AscH^−^ in its genotoxicity against NSCLC.

### 3.6. Pharmacological Ascorbate Induces Apoptosis in NSCLC through Formation of H_2_O_2_

The DNA damage by oxidative distress is strongly linked to an induction of apoptosis [54,55]. Results from flow cytometric analysis with annexin V/propidium iodide co-staining demonstrated that exposure to P-AscH^−^ caused apoptotic cell death on NSCLC cells in a dose-dependent fashion at 24 h post-treatment (Figure 7A). The percentage of cells in the early and late apoptotic stage were displayed on Figure 7B. Notably, exposure to P-AscH^−^ did not induce necrotic cell death in NSCLC cells (Figure 7A and Appendix A). The increases in NSCLC cells at the early and late stage of apoptosis due to P-AscH^−^ were completely blocked by co-treatment with extracellular catalase (Figure 7C,D). These findings strongly emphasize the essential roles of extracellular H_2_O_2_ in anti-cancer activities of P-AscH^−^ against NSCLC.

To further verify the effects of P-AscH^−^ on apoptotic induction, the key apoptotic signatures, including cleavage of poly (ADP-ribose) polymerase (PARP) and reduction in anti-apoptotic proteins (B-cell lymphoma 2, Bcl-2; and myeloid cell leukemia-1, Mcl-1), were detected at 24 h post-treatments with Western blot analysis. Treatments with P-AscH^−^ led to generation of cleaved PARP and downregulation of Bcl-2 and Mcl-1 expressions on NSCLC (Figure 8). These results strongly suggest that apoptosis could be a major mode of cell death of P-AscH^−^ against NSCLC.

### 3.7. Pharmacological Ascorbate Synergizes the Cytotoxic Effects of Chemotherapies in NSCLC Treatments

Gemcitabine and docetaxel stand as pivotal chemotherapeutic agents in the management of NSCLC therapy [56]. In our preliminary investigation, it became evident that all examined NSCLC cell lines shared a common phenotype, demonstrating comparable degrees of resistance to gemcitabine treatment, with estimated EC_50_ values exceeding 10 mM (Appendix A). Consequently, we specifically focused on H23 cells to explore the potential synergy between P-AscH^−^ and gemcitabine. Conversely, in the context of docetaxel treatment, our preliminary findings suggested that H292 cells displayed the highest resistance (Appendix A). As a result, H292 cells were selected for a more comprehensive assessment of the potential synergistic interactions of P-AscH^−^ and docetaxel.

Results from the MTT viability assay show an enhancement in the cytotoxic effects of gemcitabine when administered in conjunction with P-AscH^−^ to H23 cells (Figure 9A). Similarly, an augmentation in the anti-cancer properties of docetaxel against H292 cells were observed when combined with P-AscH^−^ (Figure 9C). Notably, the calculated combination index (CI) values for both P-AscH^−^/gemcitabine (Figure 9B) and P-AscH^−^/docetaxel (Figure 9D) consistently remained below 1. These findings strongly indicate a synergistic interplay between P-AscH^−^ and these chemotherapeutic drugs, highlighting its potential for enhanced therapeutic efficacy of conventional therapy.

## 4. Discussion

Our current investigation provides strong evidence supporting the importance of P-AscH^−^’s pro-oxidant activities in its anti-cancer properties. Our results demonstrate that P-AscH^−^ induces intracellular oxidative distress, as determined by decrease in intracellular GSH and increase in DCFH oxidation across all tested NSCLC (Figure 2). It was previously reported on other types of cancer that extracellular H_2_O_2_ plays a pivotal role in the cytotoxicity of P-AscH^−^ [5,6,7,8,9,10,11,12,13]. To assess this casual relationship on the NSCLC model, we used an extracellular catalase, H_2_O_2_ scavenger, as a tool. Co-treatment with extracellular catalase inhibited oxidative distress and rescued NSCLC from cytotoxicity of P-AscH^−^ (Figure 2). These results further emphasize the indispensability of extracellular H_2_O_2_ for anti-cancer effects of P-AscH^−^ against NSCLC.

The high-flux H_2_O_2_ generated from P-AscH^−^ oxidation can diffuse into cancer cells through peroxiporin and inflict damage on various macromolecules [57]. Due to its highly nucleophilic characteristics [58], our hypothesis is that DNA could be the main susceptible target affected by P-AscH^−^ in NSCLC treatment. Data from Western blot analysis demonstrated that P-AcH^−^ induces oxidative DNA damage via H_2_O_2_ production. Exposure to pharmacological ascorbate resulted in the generation of markers for DNA damage (γ-H2AX) and replication stress (p-Chk1 and p-RPA2) (Figure 4). Additionally, co-treatment with extracellular catalase suppressed the formation of these markers across all tested NSCLC cell lines (Figure 5). These findings strongly support the notion that production of extracellular H_2_O_2_ plays pivotal roles in the genotoxicity of P-AscH^−^ against NSCLC. The generated H_2_O_2_ produced through oxidation of P-AscH^−^ has potential to interact with ferrous ions associated with DNA via the Fenton reaction leading to the site-specific generation of highly damaging, reactive hydroxyl free radicals, and subsequently DNA damage [59].

When cells encounter genotoxic stress, there is a dramatic elevation in the utilization of bioenergetic resources by DNA repair system (e.g., hyperactivation of PARP) to rectify severe damages on DNA and to promote survival [47]. Our bioenergetics studies showed that P-AscH^−^ elicited a substantial reduction in intracellular ATP pools and NAD^+^ reserves in all NSCLC cell lines examined (Figure 3). The bioenergetics disturbances caused by P-AscH^−^ were significantly attenuated in the presence of extracellular catalase. These results emphasize the involvement of extracellular H_2_O_2_ in the induction of DNA damage (Figure 3).

Exposure to P-AscH^−^ not only targets DNA, but also disrupts the DNA repair systems of NSCLC cells. Our experimental findings, as depicted in Figure 4 and Figure 5 through Western blot analysis, show that generation of H_2_O_2_ from oxidation of P-AscH^−^ caused a reduction in overall levels of Chk1 and RPA2 in NSCLC cells. We postulated that the downregulation of these two critical components of the DNA damage response system could result in an inefficient DNA repair and ultimately trigger cell death. These findings are aligned with our observations from confocal microscopic analysis of H460 cells. We found that treatment with P-AscH^−^ caused the leakage of several vital DNA repair factors (γ-H2AX, BRCA1 and RAP80) into the cytoplasm (Figure 6). The BRCA1 and RAP80 proteins are components of the BRCA1-A complex, which regulates the repair of DNA double-strand breaks (DSBs). RAP80 helps recruit other members of BRCA1-A complex to the DSBs loci. Afterwards, colocalization of BRCA1 and γ-H2AX in the nucleus recruits other DNA damage response proteins to initiate the DNA repair processes [60,61]. Abnormal cytoplasmic accumulations of these signaling molecules are associated with an activation of cell death signals on different types of cancer, as reported in previous studies [62,63,64]. Altogether, we hypothesized that the perturbation of the key signaling molecules in the DNA damage response system following P-AscH^−^ treatment could impair the repair process of NSCLC and eventually lead to induction of cell death. Additional kinetic studies on DNA repair are required to further support this proposed mechanism.

One of the fates of cells with defects in DNA repair machineries is an induction of apoptotic cell death [65]. Our data from flow cytometric assay and Western blot analysis show that P-AscH^−^ triggers apoptotic cell death on NSCLC as determined by an increase in apoptotic cells (annexin-V positive; Figure 7); and formation of apoptotic hallmarks (cleavage of PARP; decrease in expression of anti-apoptotic Bcl-2 and Mcl-1; Figure 8), respectively. The increases in apoptotic cell death were prevented by catalase co-treatment (Figure 7). This information strongly supports the crucial roles of extracellular H_2_O_2_ in anti-cancer activities of P-AscH^−^. Notably, other modes of cell death by P-AscH^−^ were also evidenced on different models of cancer, e.g., ferroptosis (pancreatic cancer [66] and thyroid cancer [67]), and autophagy (pancreatic cancers [68], and prostate cancer [6]). The variation in features of cell deaths could possibly be due to biological and genetic characteristics of cancer cells and experimental conditions.

Pharmacological ascorbate demonstrated remarkable selectivity in targeting cancer cells, while normal cells remain unaffected by toxic outcomes attributed to P-AscH^−^ [5,7,22,23,24]. One potential rationale for this observed selectivity of P-AscH^−^ could be due to the variances in the capacity of cells to neutralize H_2_O_2_. Normal cells exhibit a greater proficiency in detoxifying extracellular H_2_O_2_ when compared to their cancerous counterparts. For instance, the *k*_cell_ value, representing the rate constant for H_2_O_2_ removal, is considerably higher in normal lung cells (*k*_cell_ of human bronchial epithelial cells ≅7 × 10^−12^ s^−1^ cell^−1^ L), whereas lung cancer cells exhibit relatively lower *k*_cell_ values ( ≅2–4 × 10^−12^ s^−1^ cell^−1^ L) [23]. Consequently, these distinct *k*_cell_ values led us to postulate that the cytotoxic effects of P-AscH^−^ in NSCLC treatments would specifically target only lung cancer cells, not normal tissues. Further in vitro and in vivo investigations are required to validate and substantiate our hypothesis regarding safety and selectivity of P-AscH^−^ in NSCLC treatments.

Pharmacological ascorbate was proposed as a potential adjunctive therapy alongside with standard-of-care (e.g., chemotherapy [25,26,27,28], radiotherapy [7,29,30], and chemoradiotherapy [31,32,33]) in management of diverse cancer types. This current study highlights DNA as a susceptible target for P-AscH^−^ treatments against NSCLC. Hence, combining P-AscH^−^ with chemotherapeutic agents that attack DNA or disrupt cell division would be a promising strategy to enhance efficacy of NSCLC treatments. Our observations on cell culture models of NSCLC provide supportive evidence for this clinical potential of P-AscH^−^. We showed that P-AscH^−^ synergizes anti-cancer effects of standard therapies of NSCLC, gemcitabine (Combination Index (CI) < 1; Figure 9B) and docetaxel (CI < 1; Figure 9D). These two chemotherapeutic agents exhibit their anti-tumor effects through distinct mechanisms; gemcitabine inhibits DNA synthesis, while docetaxel disrupts the microtubular network during mitosis. We hypothesize that the observed enhancement of anti-cancer activities when P-AscH^−^ is combined with these chemotherapeutic drugs may be attributed to an augmentation of genotoxic effects by P-AscH^−^. This hypothesis is rooted in our preliminary investigation of the presence of γ-H2AX in the combination regimen, as discussed in Appendix A. To further substantiate this hypothesis, comprehensive assessments using quantitative approaches to quantify the damage on DNA are essential. Taken together, incorporating P-AscH^−^ into standard-of-care therapies holds significant promise as a strategy to improve the effectiveness of standard regimens and enhance the quality of life for individuals with cancer.

## 5. Conclusions

In summary, our investigation provides compelling evidence for the clinical promise of P-AscH^−^ in NSCLC treatment. We uncovered novel and valuable insights into the biochemical mechanisms underlying the anticancer actions of P-AscH^−^, with a particular emphasis on DNA damage as a significant contributor to the therapeutic efficacy of P-AscH^−^. Our research demonstrated that P-AscH^−^ induces oxidative distress, disrupts cellular bioenergetics homeostasis, triggers apoptotic cell death and reduces clonogenic survival. Importantly, we identified DNA and DNA damage response cascades as vulnerable targets for P-AscH^−^ in NSCLC therapy. The observed elevation of DNA damage and replication stress markers, alongside the mislocalization of DNA repair machineries, underscores the genotoxic and cytotoxic effects of P-AscH^−^. Notably, the reversal of these effects by extracellular catalase emphasizes the roles of extracellular H_2_O_2_ in mediating P-AscH^−^‘s anti-cancer actions. We firmly believe that the insights gained from our current research will advance our understanding of P-AscH^−^ in cancer treatment and support its potential as a clinical therapeutic option for combating against NSCLC.

## Figures and Tables

**Figure 1 antioxidants-12-01775-f001:**
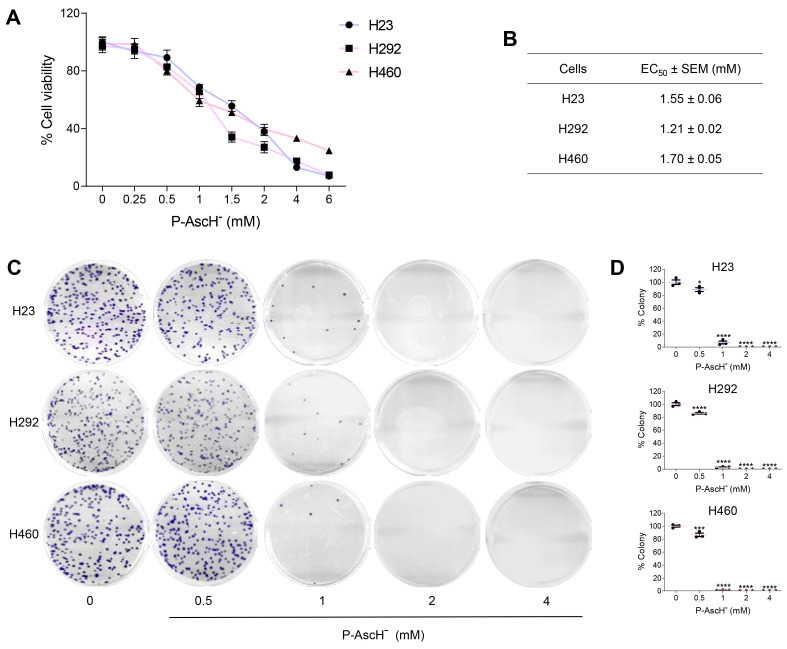
Pharmacological ascorbate possesses anti-cancer activities against NSCLC cells. (**A**) P-AscH^−^ kills NSCLC cells in dose-dependent fashion. Cells were exposed to P-AscH^−^ (0–6 mM; 1 h) and cell viability was evaluated at 24 h after treatments using MTT assay. (**B**) The concentration of P-AscH^−^ required to decrease viability of H23, H292, or H460 by 50% (EC_50_) was further evaluated on the basis of the dose–response relationship. (**C**,**D**) P-AscH^−^ dose-dependently reduces clonogenicity of NSCLC cells. Cells were treated with P-AscH^−^ (0–4 mM; 1 h) and the ability of cancer cells to form a colony was determined following treatments with clonogenic survival assay (*n* = 3; mean ± SEM; * *p* < 0.05; *** *p* < 0.001; **** *p* < 0.0001 vs. untreated control).

**Figure 2 antioxidants-12-01775-f002:**
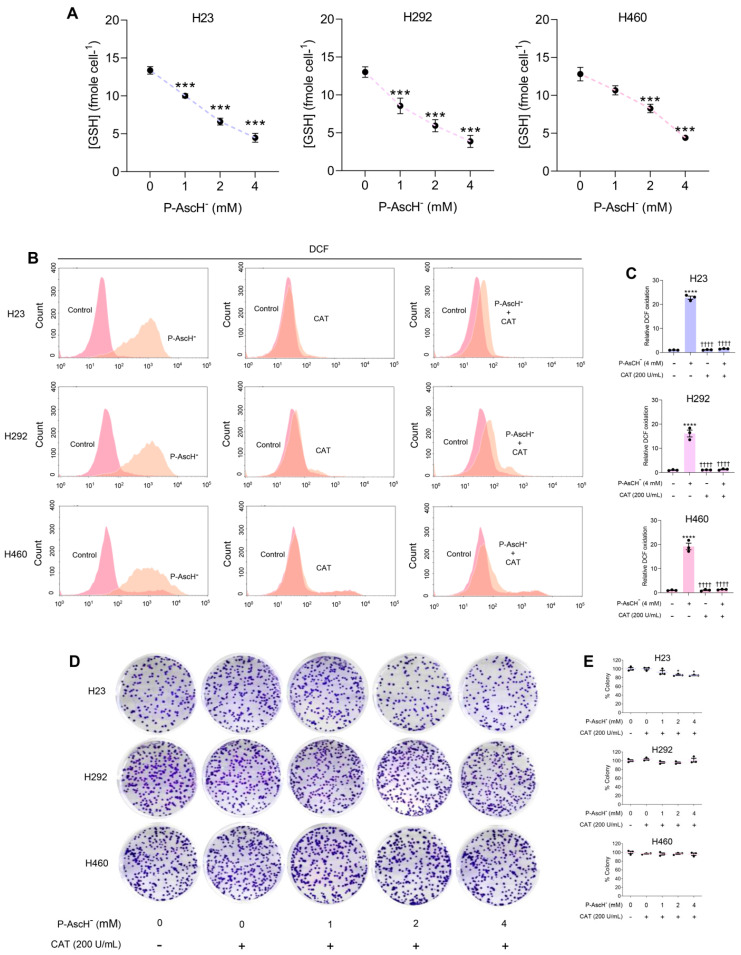
The generation of extracellular H_2_O_2_ is an essential factor for anti-cancer properties of P-AscH^−^ against NSCLC cells. (**A**) The intracellular levels of GSH of NSCLC cells were reduced after P-AscH^−^ treatments. Cells were exposed to P-AscH^−^ (0–4 mM) for 1 h; and the redox status of NSCLC cells was immediately evaluated by a measurement of an intracellular concentration of GSH. (**B**,**C**) P-AscH^−^ induces generation of oxidative distress in NSCLC cells via the formation of H_2_O_2_. To investigate roles of H_2_O_2_, NSCLC cells were treated with P-AscH^−^ (4 mM) ± extracellular catalase (200 U/mL) for 1 h. The oxidative status of cells was subsequently determined by flow cytometric analysis using DCFH-DA fluorescent probe. (**D**,**E**) The formation of extracellular H_2_O_2_ is crucial for cytotoxicity of P-AscH^−^ against NSCLC cells. Co-treatment with catalase prevented a decrease in clonogenic survival of NSCLC cells due to P-AscH^−^ exposure. The treatment protocols were similar to those described in (**B**,**C**) (*n* = 3; mean ± SEM; * *p* < 0.01, *** *p* < 0.001; **** *p* < 0.0001 vs. untreated control; ^††††^ *p* < 0.0001 vs. P-AscH^−^).

**Figure 3 antioxidants-12-01775-f003:**
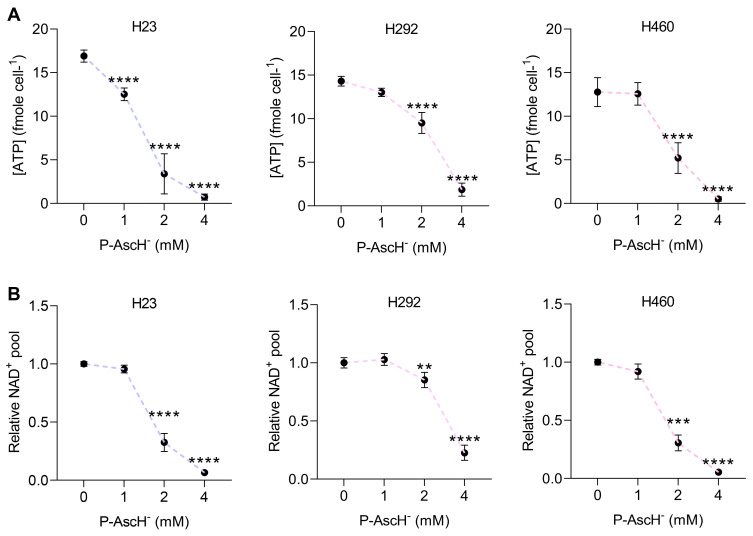
Pharmacological ascorbate exhausts intracellular storage of ATP and NAD^+^ via generation of H_2_O_2_. (**A**,**B**) The intracellular pools of ATP and NAD^+^ of NSCLC cells were rapidly depleted upon treatment with P-AscH^−^. Cells were incubated with P-AscH^−^ (0–4 mM) for 1 h, and the immediate impact on intracellular amounts of ATP (**A**) and NAD^+^ pool (**B**) was observed. (**C**,**D**) The disruption of cellular bioenergetics by P-AscH^−^ is primarily mediated by formation of extracellular H_2_O_2_. Catalase prevents the reduction in ATP levels (**C**) and NAD^+^ storage (**D**) in NSCLC cells following P-AscH^−^ treatment. The treatment protocols were similar to 2B and 2C (*n* = 3; mean ± SEM; ** *p* < 0.01, *** *p* < 0.001, **** *p* < 0.0001 vs. untreated control; ^††††^ *p* < 0.0001 vs. P-AscH^−^).

**Figure 4 antioxidants-12-01775-f004:**
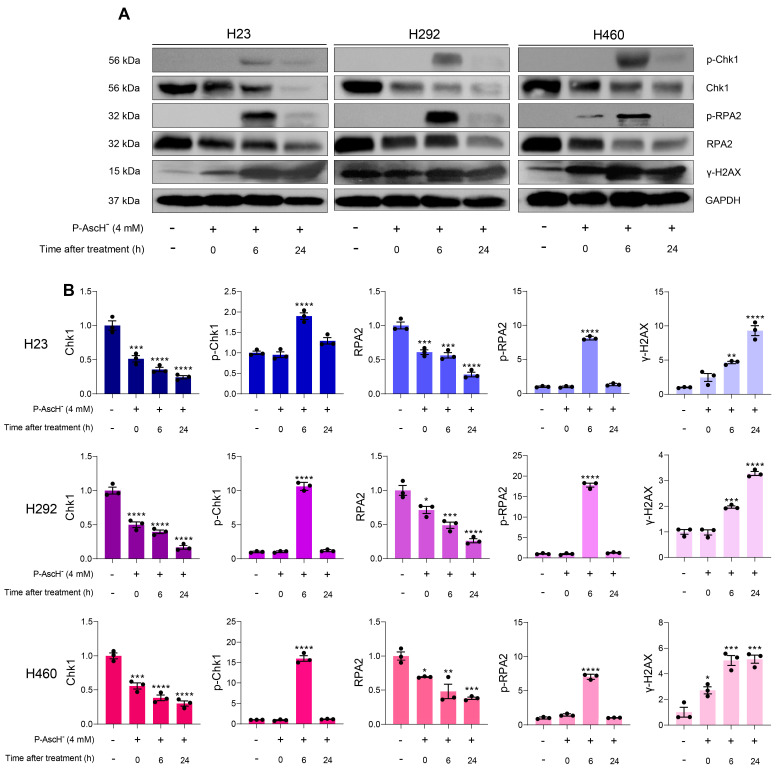
DNA is a susceptible target for treatment of P-AscH^−^. (**A**) Representative data from Western blot analyses demonstrate that P-AscH^−^ treatment leads to a decrease in DNA damage regulators (Chk1 and RPA2); as well as an increase in markers of DNA damage and replication stress following P-AscH^−^ exposure (phosphorylation at Ser324 of Chk1, p-Chk1; phosphorylation at Ser33 of RPA2, p-RPA2; phosphorylation at Ser139 of histone H2AX, γ-H2AX). Cells were treated with P-AscH^−^ (4 mM; 1 h) and the alterations in DNA damage response system were subsequently determined with Western blot assays at indicated time points (0–24 h after treatments). (**B**), The relative expression of the indicated proteins compared to the untreated control were calculated from densitometric analysis and normalized to GAPDH expression (*n* = 3; mean ± SEM; * *p* < 0.05, ** *p* < 0.01, *** *p* < 0.001, **** *p* < 0.0001 vs. untreated control).

**Figure 5 antioxidants-12-01775-f005:**
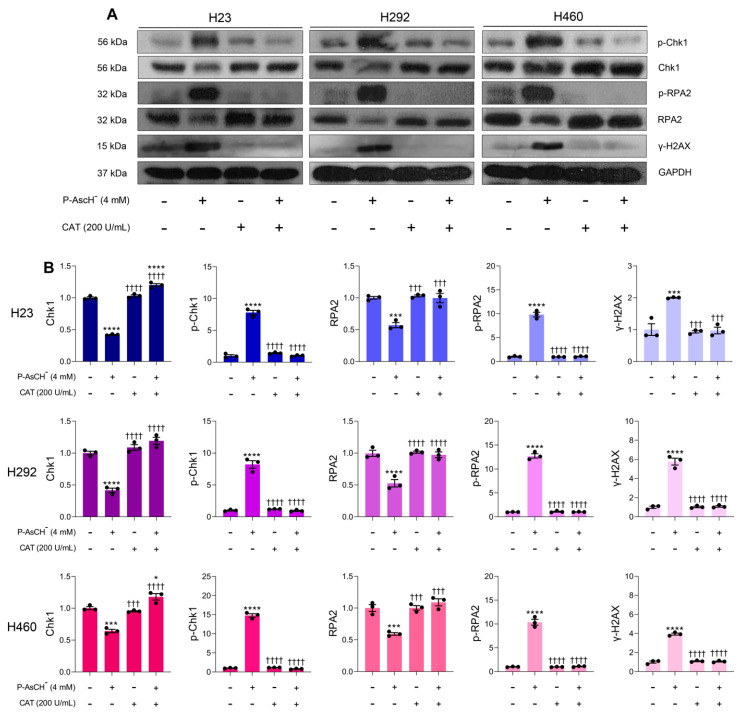
Formation of H_2_O_2_ is an essential factor for genotoxicity of P-AscH^−^ against NSCLC. (**A**) Catalase prevents loss of Chk1 and RPA2; and inhibits formation of p-Chk1, p-RPA2, and γ-H2AX. Cells were co-treated with P-AscH^−^ (4 mM) and bovine catalase (200 U/mL) for 1 h, then the changes in DNA damage response pathway were determined at 6 h post-treatment with Western blot analyses. Data are representative of three independent studies. (**B**) The density of each blot was estimated and subsequently normalized to loading control GAPDH. The values were expressed as fold change relative to the untreated control (*n* = 3; mean ± SEM; * *p* < 0.05, *** *p* < 0.001, **** *p* < 0.0001 vs. untreated control; ^†††^ *p* < 0.001, ^††††^ *p* < 0.0001 vs. P-AscH^−^).

**Figure 6 antioxidants-12-01775-f006:**
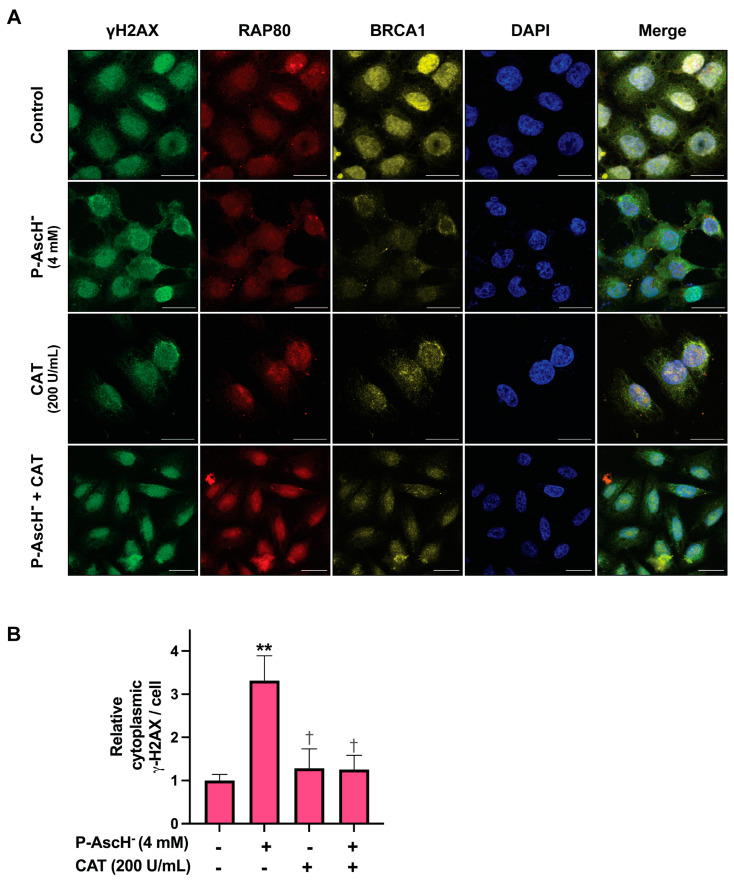
Pharmacological ascorbate affects the localization of key components of the DNA damage response pathway in NSCLC cells. (**A**) Immunofluorescence staining: H460 cells were treated with 4 mM P-AscH^−^, bovine catalase (200 U/mL), or co-treated with P-AscH^−^ (4 mM) and catalase (200 U/mL) for 1 h. Then, cells were subjected to immunofluorescence staining with anti-γ-H2AX (green), anti-RAP80 (red), and anti-BRCA1 (yellow) antibodies at 6 h post-treatment to observe protein localization. Cell nuclei were stained with DAPI (blue). Data are representative of three independent studies. Scale bar, 20 µm. (**B**) The levels of cytoplasmic γ-H2AX per cell were quantified. The data were expressed as fold change relative to the untreated control (*n* = 3; mean ± SEM; ** *p* < 0.01 vs. untreated control; ^†^ *p* < 0.05 vs. P-AscH^−^).

**Figure 7 antioxidants-12-01775-f007:**
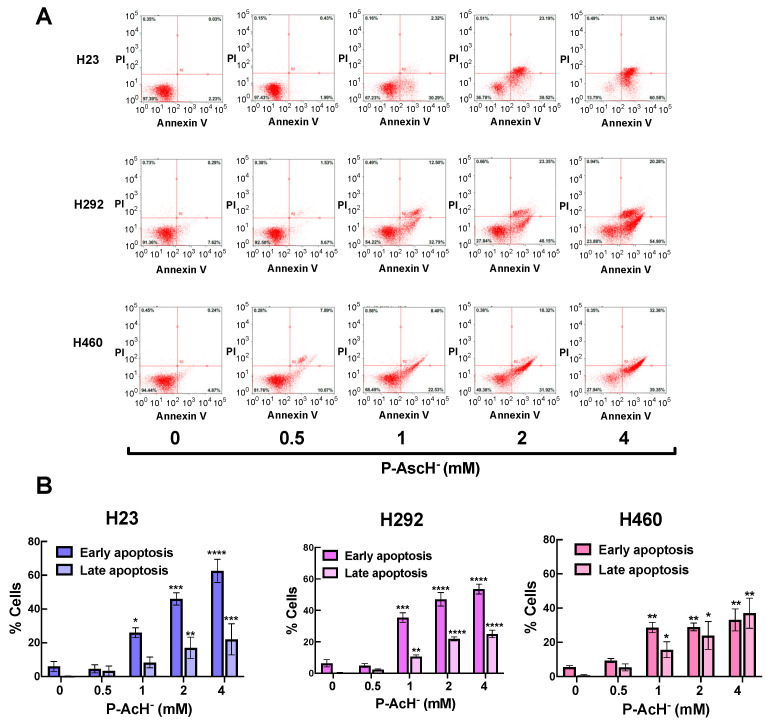
Pharmacological ascorbate causes apoptotic cell death in NSCLC via H_2_O_2_ production. (**A**,**B**) Exposure to P-AscH^−^ (1–4 mM; 1 h) induces apoptotic cell death in NSCLC in a dose-dependent fashion. (**C**,**D**) The inductions of apoptosis by P-AscH^−^ (1–4 mM; 1 h) on NSCLC were prevented by catalase co-treatment. Cells were treated with P-AscH^−^ (0–4 mM) ± catalase (200 U/mL) for 1 h. At 24 h after treatments, the modes of cell death were characterized with flow cytometric assay using annexin V/PI co-staining (*n* = 3; mean ± SEM; * *p* < 0.05, ** *p* < 0.01; *** *p* < 0.001, **** *p* < 0.0001 vs. untreated control).

**Figure 8 antioxidants-12-01775-f008:**
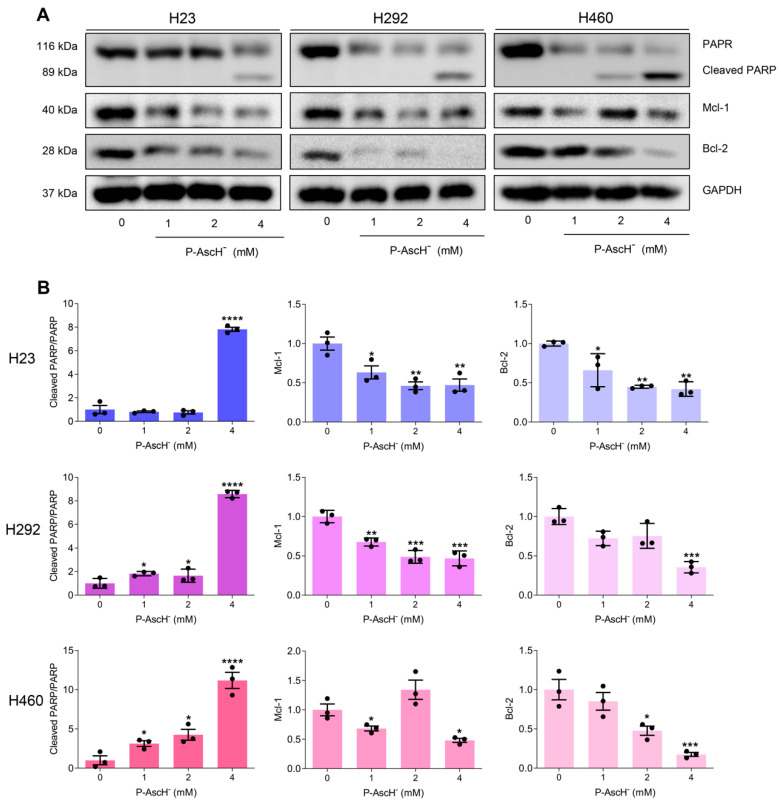
Pharmacological ascorbate induces activation of apoptotic signaling cascade on NSCLC. (**A**) The hallmarks of apoptotic cell death (increased PARP cleavage and reduced expression of anti-apoptotic proteins (Mcl-1 and Bcl-2)) were observed in NSCLC with Western blot analysis at 24 h after treatments with P-AscH^−^ (0–4 mM; 1 h). (**B**) The density of each protein band was measured and normalized to GAPDH. The data were presented as fold change relative to the untreated control (*n* = 3; mean ± SEM; * *p* < 0.05, ** *p* < 0.01; *** *p* < 0.001, **** *p* < 0.0001 vs. untreated control).

**Figure 9 antioxidants-12-01775-f009:**
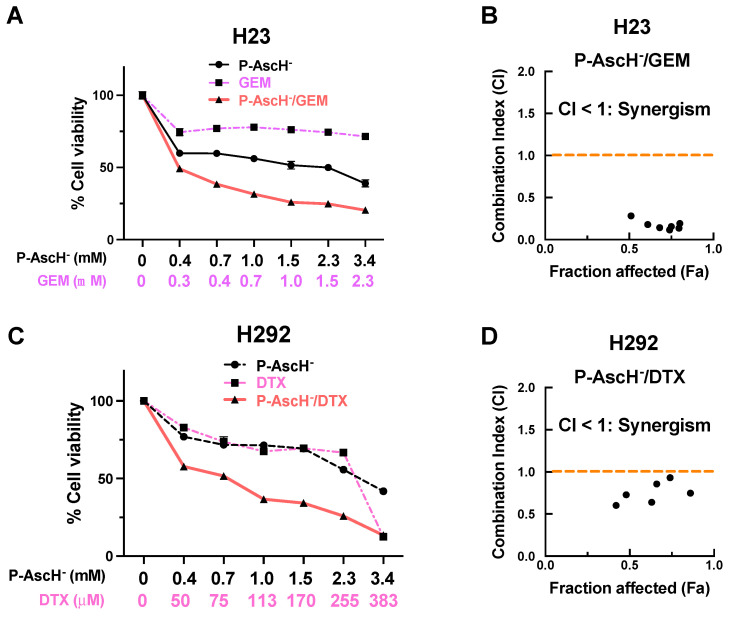
Pharmacological ascorbate synergistically enhances anti-cancer activities of chemotherapeutic agents against NSCLC. (**A**) The addition of P-AscH^−^ augments the cytotoxicity of gemcitabine (GEM) in H23 cells. (**B**) The graph illustrates the combination indices (CI) plotted against the fraction affected (Fa), indicating the synergistic interaction between P-AscH^−^ and GEM. (**C**) P-AscH^−^ amplifies the anti-cancer activities of docetaxel (DTX) in H292 cells. (**D**) The CI values between P-AscH^−^/DTX are consistently below 1, suggesting the synergistic effects of this combination. Cells were exposed to varying amounts of chemotherapeutic drugs (either GEM or DTX; 24 h); or P-AscH^−^ (1 h); or a combined treatment of respective chemotherapeutic drugs (24 h) followed by P-AscH^−^ (1 h). At 24 h post-treatments, cell viability was assessed with MTT assay and combination indices were calculated to assess interaction effect. For combination treatments, the consistent ratio of GEM:P-P-AscH^−^ (1:1.5) and DTX: P-AscH^−^ (1:8.9) were maintained, respectively. The presented data were derived from three independent experiments.

## Data Availability

Data is contained within the article and Appendix A.

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
