# Peer review of "Pharmacological Ascorbate Elicits Anti-Cancer Activities against Non-Small Cell Lung Cancer through Hydrogen-Peroxide-Induced-DNA-Damage"

_antioxidants, 2023, doi:10.3390/antiox12091775_

Round 1
Reviewer 1 Report
Antioxidants 2555062 -peer review report
General considerations
First, the reviewer wishes to express their gratitude to the editors for giving them the opportunity to help the authors and to the authors for sharing their research.
The manuscript deals with the use of pharmacological amounts of ascorbic acid as an anti-cancer agent. While the effects of this treatment have been studied for other tumours, the authors identified a gap in the literature regarding NSCLC and here describe the effects of P-AscH on three NSCLC cell lines in terms of oxidation, changes in bioenergetics, survival and DNA damage. The authors identify a high extracellular H2O2 flux as the principal mechanism of action of P-AscH by the use of extracellular catalase and report strong effects on DNA damage and the induction of apoptosis.
The reviewer thanks the authors for including uncropped pictures of all their western blots in the supplementary material and for adding brief explanations of the rationale and method of detection of the bioenergetics and oxidation assays used in the manuscript. The reviewer believes this is a plus for readers who might come from a different field or for less experienced readers. The reviewer was also glad that the authors used two independent assays to quantify oxidation in cells and demonstrated awareness of the weaknesses and strengths of different approaches.
Issues to address and clarification requests.
1. Cell line choice – It would be useful to know why the three cell lines were chosen (plus A549). Did the authors choose them to represent different subtypes of NSCLC (e.g. EGFR mutated, KRAS mutated, etc)? Can the authors clarify?
2. No normal lung epithelial cell line – the literature on P-AscH suggests that the oxidative attack of the treatment is so damaging to cancer cells because they lack the oxidation defense mechanisms necessary to detoxify extracellular H2O2, whereas normal cells would be comparatively spared because they have functional antioxidant mechanisms. I fell the authors missed an opportunity of checking that this holds true for normal lung epithelium by not including such a cell line at least at the beginning of the study. I am well aware of the difficulty of working with primary cell lines and the like, so I am not asking that the authors do so at such a late date, but perhaps it bears thinking for a follow-up work.
3. Confocal data and its interpretation – it is clear from references 57-59 that the accumulation of DNA damage repair proteins in the cytoplasm of cancer cells is indicative of cytotoxicity and might be a cause of cytotoxicity. In Figure 6 the authors make the case that this is what happens in the 3 cell lines in this study, however the result is left as a qualitative observation, which the size of the figure on a screen and the quality of a printout make rather hard to judge accurately. I would request that the authors make an attempt to quantify this effect using ImageJ or another image analysis software and display the results with a statistical test.
Additionally, it would be great to have the parameters for laser power, gain and filters used for the experiment. They appear to be missing from M&M section 2.13.
4. Supplementary figure S5 clarifications – the data in figure S5 is rather interesting and I wonder why it’s been relegated to Supplementary Materials instead of being included in the main body of the work. However, I fell that displaying only the CI plot is a bit reductive and I would ask the authors to add some panels with the original MTT data used to calculate the CI.
Also the hypothesis that P-AscH might augment the genotoxic effect of gemtabicin and docetaxel, respectively, is very interesting, and seems to be supported by the literature for other cancer types. In this view, it is kind of disappointing that the authors haven’t shown any data to this effect. If the authors have it, I would strongly suggest that they add it to the manuscript – or even that they test it from scratch if practically possible.
Finally, I would like to ask why each drug was tested in a single cell line and how the cell line was chosen for this.
5. Missing reference to Figure S3 in section #3.2 - I think it would be worth mentioning it when discussing comparable results in Fig 3A and 3B.
6. Reference missing for Chk1 rationale – based on the structure of the preceding paragraph about RPA, I was expecting to see a reference for the biology of Chk1 at line 393. Perhaps it is missing?
7. Manufacturer missing for materials in section 2.1 lines 99-104
8. RRID for cell lines – would it be possible to check it and add it for disambiguation purposes? The ATCC accession code would also work.
9. A549 cell line not mentioned in M&M - please add it
10. The product code and concentration of the protease and phosphatase inhibitor cocktail is missing - please add it.
11. Reference #54 is a duplicate of #43 – please correct it.
12. A few typos throughout the manuscript.
Reviewer 2 Report
In this study, the authors examined the anti-cancer effects of P-AscH- on NSCLC and its mechanisms. They showed that P-AscH- treatment induced cellular oxidative distress; disrupts cellular bioenergetics; and led to induction of apoptotic cell death and reduction in clonogenic survival. DNA damage response machineries were activated by P-AscH-. Treatments with P-AscH- increased the formation of DNA damage and replication stress markers as well as mislocalization of DNA repair machineries. The cytotoxic and genotoxic effects of P-AscH- on NSCLC were prevented by exogenous catalase, indicating the roles of hydrogen peroxide in anti-cancer activities of P-AscH-. The authors concluded that their findings support P-AscH- potential clinical use as a therapeutic option for NSCLC therapy. The specific comments for this manuscript are listed below.
Major
1. The major concern of this study is that nearly all of the results were reproduced experiments of published reports (references #5-13). The only difference is that this study using NSCLC cell lines as research models. Although the results are still informative, the authors failed to emphasize what was the important finding in their study.
2. EGFR mutations (as well as K-Ras mutation) are frequently occurred in lung adenocarcinoma (LAC). If the authors used LAC cell lines with different oncogenic mutations and showed how they respond to P-AscH-, the data will provide novelties and this study may become more informative and significant.
3. P-AscH-induced hydrogen peroxide may be formed in the extracellular compartment, but it can enter the cells easily and cause all the downstream effects. The emphasis on extracellular location of P-AscH-induced hydrogen peroxide is unnecessary.
Minor
4. During drug or treatment development processes, preclinical development or preclinical studies is a stage of research that begins before clinical trials (testing in humans); during which important feasibility, iterative testing and drug safety data are collected and examined, usually in laboratory animals. This study cannot be stated as a preclinical study and the authors are advised to avoid using the term “preclinical” to described their study.
5. The text between lines 210-215 can be deleted. It is not necessary to described the principle of the NAD+ pool measurement. Only the material and method need to be explained.
none
Round 2
Reviewer 2 Report
The revision has addressed previous concerns. No additional question.